# Genome-Wide Identification, Expression, and Interaction Analysis of the Auxin Response Factor and *AUX*/*IAA* Gene Families in *Vaccinium bracteatum*

**DOI:** 10.3390/ijms25158385

**Published:** 2024-08-01

**Authors:** Xuan Gao, Xiaohui Liu, Hong Zhang, Li Cheng, Xingliang Wang, Cheng Zhen, Haijing Du, Yufei Chen, Hongmei Yu, Bo Zhu, Jiaxin Xiao

**Affiliations:** 1Anhui Provincial Key Laboratory of Molecular Enzymology and Mechanism of Major Metabolic Diseases, College of Life Sciences, Anhui Normal University, Wuhu 241000, China; gaoxuan@ahnu.edu.cn (X.G.); liuxiaohui@ahnu.edu.cn (X.L.); cl17855368613@163.com (L.C.); c2320288389@163.com (C.Z.); 18355632985@163.com (H.D.); 2Anhui Provincial Engineering Research Centre for Molecular Detection and Diagnostics, College of Life Sciences, Anhui Normal University, Wuhu 241000, China; 18672312335@163.com (H.Z.); wangxingliang2000@163.com (X.W.); cyf2130035271@163.com (Y.C.); yhm1201@ahnu.edu.cn (H.Y.)

**Keywords:** auxin response factor, cis-elements, domain analysis; expression profiling

## Abstract

Background: Auxin, a plant hormone, plays diverse roles in the modulation of plant growth and development. The transport and signal transduction of auxin are regulated by various factors involved in shaping plant morphology and responding to external environmental conditions. The auxin signal transduction is primarily governed by the following two gene families: the auxin response factor (*ARF*) and auxin/indole-3-acetic acid (*AUX*/*IAA*). However, a comprehensive genomic analysis involving the expression profiles, structures, and functional features of the *ARF* and *AUX*/*IAA* gene families in *Vaccinium bracteatum* has not been carried out to date. Results: Through the acquisition of genomic and expression data, coupled with an analysis using online tools, two gene family members were identified. This groundwork provides a distinguishing characterization of the chosen gene families in terms of expression, interaction, and response in the growth and development of plant fruits. In our genome-wide search of the *VaARF* and *VaIAA* genes in *Vaccinium bracteatum*, we identified 26 *VaARF* and 17 *VaIAA* genes. We analyzed the sequence and structural characteristics of these *VaARF* and *VaIAA* genes. We found that 26 *VaARF* and 17 *VaIAA* genes were divided into six subfamilies. Based on protein interaction predictions, *VaIAA1* and *VaIAA20* were designated core members of *VaIAA* gene families. Moreover, an analysis of expression patterns showed that 14 *ARF* genes and 12 *IAA* genes exhibited significantly varied expressions during fruit development. Conclusion: Two key genes, namely, *VaIAA1* and *VaIAA20*, belonging to a gene family, play a potentially crucial role in fruit development through 26 *VaARF*-*IAA*s. This study provides a valuable reference for investigating the molecular mechanism of fruit development and lays the foundation for further research on *Vaccinium bracteatum*.

## 1. Introduction

Auxin is a pivotal hormone in plants that is crucial for the regulation of growth and development across various stages of the plant lifecycle [1]. It plays a central role in numerous biological processes including the development of various organs, regulation of shoot apical dominance, and differentiation of conducting tissues like xylem and phloem [2]. In *Arabidopsis*, *YUCCA2 (YUC2)* regulates chloroplast RNA editing by modulating auxin levels, affecting the expression of editing factors through ARF1-dependent pathways [3]. Genome-wide analysis in *Mikania micrantha* identified 18 *YUCCA* genes with diverse expression patterns and significant roles in plant growth and auxin biosynthesis [4]. Auxin also modulates responses to both biotic and abiotic stresses, making its signaling pathways critical targets for agricultural innovation [5].

At the heart of auxin signaling are the following two key families of transcription factors: the auxin response factor (*ARF*) family and the *AUX/IAA* repressor family [6]. These proteins are essential for the transcriptional regulation of auxin-responsive genes. *ARF*s recognize auxin-responsive elements (AuxREs) in the promoter regions of target genes, which typically include a TGTCTC motif or its variations [7]. Structurally, *ARF*s are characterized by a DNA-binding domain (DBD) incorporating a B3-like DNA-binding domain [8], a middle region (MR) that acts either as a repression or an activation domain, and a carboxyl-terminal domain (CTD) that mediates oligomerization with other *ARF*s or AUX/IAA proteins [9]. The modulation of *ARF* activity is intricately linked to their interaction with AUX/IAA proteins. In the absence of auxin, AUX/IAA proteins bind to *ARF*s, inhibiting the expression of auxin-responsive genes. Upon auxin binding, AUX/IAA proteins undergo ubiquitination and subsequent degradation via the 26S proteasome, thereby freeing *ARF*s to activate their target genes [1,10].

In the two decades since the identification of the inaugural member of the auxin response factor (*ARF*) family, 23 *ARF* members have been identified and characterized within the genome of *Arabidopsis thaliana* (L.). With the advancements in whole-genome sequencing, researchers have been able to ascertain the presence of 22 [11], 25 [12], 23 [13], 25 [14], and 36 [15] *ARF* genes within the genomic structures of *Solanum lycopersicum* L., *Oryza sativa*, *Triticum aestivum* L., *Sorghum bicolor* (L.) *Moench*, and *Zea mays* L., respectively. It is noteworthy that reports elucidating certain biological functionalities associated with *ARF*s have also begun to garner attention. An interesting aspect of these *ARF* genes is their predilection to perform functions in different plants, offering a fascinating dimension to their biological capacities.

Previous studies have highlighted the complexity of auxin signaling, noting that the specificity of *ARF* and AUX/IAA interactions can lead to diverse developmental and physiological outcomes [16]. The cell-type specificity of *ARF*-*AUX/IAA* expression in relation to different local auxin sources is a critical area of study in plant development. Research highlights how distinct cell types interpret auxin signals to mediate various developmental processes. Rademacher [17] provided a cellular expression map of *ARF* genes, showing complex overlapping patterns during embryogenesis and primary root meristem development. The study underscores the developmental specificity generated by the transcriptional regulation of *ARF* genes. Bargmann and Estelle [18] speculated on the tissue specificity of auxin responses through the SCF(TIR1)-*AUX/IAA*-*ARF* pathway, addressing how different cell types respond to the same auxin gradient. Ding [19] mapped the tissue-specific contributions of auxin signals in root growth, demonstrating significant roles of specific cell types like the endodermis. Specific *ARF*s regulate processes ranging from leaf senescence and floral organ abscission to vascular patterning and reproductive development in model plants such as *Arabidopsis thaliana* [3,20]. This underscores the potential of similar genes in *Vaccinium bracteatum* to affect key traits such as fruit development and stress tolerance [21].

*ARF*s and other TF families such as WRKY and MADS have recently been defined as instrumental in regulating various auxin-mediated responses [22]. Functional characterization of *AUX/IAA*-encoding genes has been testified through mutant and over-expressed studies conducted in *Arabidopsis thaliana* (L.). For instance, loss-of-function mutations in *AtIAA3* and *AtIAA28* resulted in lateral root formation defects [23,24]. It is discerned that *AtIAA12* and *AtIAA18* [25] can modulate embryonic apical configuration and root meristem formation by inhibiting *AtARF5* [26] activity. In *Arabidopsis*, the *AtARF7* gene participates in water absorption and root growth and, similarly [27], maize genes *ZmARF25* and *ZmARF34* [28] play significant roles in the regulation of lateral and crown root development. In the case of tomatoes, *ARF* genes occur as pivotal players in controlling chloroplast development and the accumulation of sugar and carotenoids, as well as fruit ripening. Simultaneously, *SlARF2A* and *SlARF2B* are also implicated in fruit ripening regulation. Upon silencing of *SlARF2A*, *SlARF2B* [5], or both, fruit ripening is severely inhibited and cannot be restored by exogenous ethylene treatment. Furthermore, *SlARF2* is involved in regulating the senescence of tomato floral organs. In rice, the *OsARF11* and *OsARF15* [29] genes, and in kiwi, the *AcARF4*, *AcARF5*, *AcARF23a*, and *AcARF28a* genes are significantly upregulated after being induced by salt stress.

Importantly, *Vaccinium bracteatum Thunb.*, an Ericaceae plant [30], is known for its berries that turn purple-black when ripe, bearing flowers in June–July and fruits in August–October. These small, deep blue fruits are naturally high in anthocyanin content, making this species a valuable ornamental fruit-greening plant. The plant’s leaves contain a substantial amount of plant polyphenols, which bestow high dietary and health preservation value. In *Vaccinium bracteatum* [31], a species renowned for its medicinal and nutritional value [32], understanding the role and regulation of *ARF* and IAA genes could provide significant insights into its growth and stress response mechanisms [30]. This study aims to provide a comprehensive analysis of the *ARF* and IAA gene families in *Vaccinium bracteatum*, exploring their gene structures, expression profiles, and evolutionary relationships within and across species [20]. By elucidating the phylogenetic relationships and functional annotations of the *ARF* and IAA genes in *Vaccinium bracteatum*, this study aims to contribute to the broader understanding of auxin-mediated regulation in plants [33], potentially guiding future agricultural strategies to exploit these pathways for crop improvement [34,35,36].

## 2. Results

### 2.1. Identification of VaARF-IAA Families in Vaccinium bracteatum

Through a meticulous HMM analysis followed by BLASTP comparison against the NCBI database, we successfully identified a total of 26 *VaARF* and 17 *VaIAA* genes in *Vaccinium ashei* (*Vaccinium bracteatum*) (Appendix A), representing a comprehensive catalog of genes implicated in the auxin response and signaling pathways. The open reading frames of these *VaARF*-IAA genes range from 173 to 531 amino acids in length, reflecting a significant diversity in protein size and potential function within the family. The predicted isoelectric points (PIs) of the resultant proteins span from 5.45 to 9.08, with molecular weights (MWs) ranging from 22,120.92 to 133,143.06 Da, highlighting the biochemical diversity of these auxin response factors and auxin-induced proteins in *Vaccinium bracteatum*.

### 2.2. Chromosomal Localization and Collinearity Analysis of VaARF-IAA Gene Families in Vaccinium bracteatum

#### 2.2.1. Chromosomal Distribution and Gene Duplication Events

Chromosome 9 has the highest density of *VaARF* genes, with a total of five identified, whereas chromosomes 1, 3, 4, 5, and 10 each harbor a single *VaARF* gene (Figure 1A). The *VaIAA* genes exhibit a similarly uneven distribution, with chromosome 5 containing the most *VaIAA* genes (four), and chromosomes 2 and 9 showing no *VaIAA* gene presence. Further analysis revealed several instances of gene clustering, suggesting potential gene duplication events. This distribution pattern suggests a non-uniform duplication and divergence history across different chromosomes, possibly influenced by specific evolutionary pressures or chromosomal environments.

#### 2.2.2. Synteny Relationships Highlighting Evolutionary Conservation

The collinearity analysis of *ARF* and *IAA* gene families in *Vaccinium bracteatum* was conducted using TBtools (version 2.019) software, focusing on assessing the chromosomal distribution and potential gene duplication events within the genome. Gene families can be generated by tandem and segmental duplications of chromosomal regions as well as whole-genome duplications (WGDs) [37]. Two homologous repeats within the same chromosome that are less than 200 kb apart are usually referred to as tandem repeats [38]. A significant finding from this study was the observation of tandem duplications and segmental duplications that contribute to the expansion of the *ARF* and *IAA* gene families. For instance, within chromosome 9, two pairs of *VaARF*s (*VaARF19-2* and *VaARF19-4*; *VaARF31-1* and *VaARF31-2*) were identified, spaced 137 kb and 18 kb apart, respectively. The nucleotide homology between *VaARF19-2* and *VaARF19-4* was calculated at 68.0%, and between *VaARF31-1* and *VaARF31-2* at 85.7%, as detailed in Appendix A. Such closely linked gene pairs underscore the role of tandem duplications in the local expansion of gene families within the genome.

Furthermore, the analysis also highlighted several instances of potential segmental or whole-genome duplications. For example, three pairs of *ARF* genes, two pairs of *IAA* genes, and additional clusters involving four *ARF* genes and a multi-gene family of *IAA* showed homology, suggesting their origin from more extensive duplication events. The non-synonymous to synonymous substitution ratio (Ka/Ks) for these gene pairs was calculated and found to be less than 1 for all instances, as shown in Appendix A. This indicates that the duplicated gene pairs and clusters have undergone purifying selection, which typically restricts functional diversification and preserves essential functions.

### 2.3. Phylogenetic Analysis of VaARF-IAA Gene Families in Vaccinium bracteatum

The phylogenetic relationships among auxin response factor (*ARF*) genes from *Vaccinium bracteatum* and other species were elucidated through comprehensive analyses utilizing ClustalX for multiple sequence alignment and PhyML software for phylogenetic tree construction via the maximum likelihood method. This analysis encompassed *ARF* nucleotide sequences from five species (Appendix A) including *Vaccinium bracteatum* (26 *VaARF*s), *Arabidopsis thaliana* (21 AtARFs), *Glycine max* (37 GmARFs), *Brassica napus* (27 BnARFs), and *Gossypium hirsutum* L. (37 GhARFs), totaling 146 *ARF* genes (Figure 2). The resultant phylogenetic tree distributed these 146 *ARF* genes into six distinct subgroups (I–VI), illustrating the evolutionary breadth of the *ARF* gene family across these species. Notably, *VaARF1-1* showed close evolutionary proximity to *ARF* genes from the other four examined species, while *VaARF22* shared a tight phylogenetic relationship with *AtARF3*, highlighting instances of conserved evolutionary trajectories. Additionally, 141 *IAA* genes were categorized into six subgroups (I–VI), including 18 *VaIAA*s, 24 *AtIAAs*, 45 *GmIAAs*, 31 *BnIAAs*, and 33 *GhIAAs*. Within this framework, *VaIAA1* was closely related to *GhIAA14*, and *VaIAA27-3* was near *GhIAA27*, underscoring the homology and potential functional conservation across species.

### 2.4. Gene Structure and Conserved Domain Analysis of VaARF-IAA Gene Families in Vaccinium bracteatum

The majority of *VaARF* genes are characterized by more than five introns, highlighting the intricate architecture of these genes. Specifically, *VaARF14-1*, *VaARF14-2*, and *VaARF30-2* are unique in their structure, each comprising a single exon, devoid of any introns (Figure 3). In contrast, *VaARF17*, *VaARF19-3*, and *VaARF31-1* contain two introns each. Similarly, the majority of *VaIAA* genes possess around five introns, with *VaIAA33* and *VaIAA27-2* featuring one and two introns, respectively. This variability underscores the evolutionary diversification within the *ARF* and *IAA* gene families.

Upon domain analysis, all identified *VaARF*s were found to contain the conserved B3 domain (Figure 4A,B), and most harbored the PB1 domain, affirming the conservation of these domains within the *VaARF* family. These domains are likely crucial for the functional repertoire of the *ARF* genes, with the B3 domain involved in DNA binding and the PB1 domain involved in protein–protein interactions. Interestingly, *VaARF14* encompasses an AP2_ERF domain, and *VaARF3* uniquely possesses the KS3_2 domain, hinting at specialized functions beyond typical *ARF* gene roles. For the *VaIAA* gene family, except for *VaIAA17*, which is composed of three B3 domains, the rest predominantly feature the PB1 domain, further illustrating the functional conservation and diversification within these auxin-response gene families.

Based on the protein sequences of *VaARF*s and *VaIAA*s, these conserved motifs were discovered using the MEME website. The conserved motifs of VaARF and VaIAA proteins were analyzed, and 12 conserved motifs (motifs 1–12) were found in both proteins (Figure 5A,B). Their sequences are shown on the right side of Figure 5. In addition to motifs 8 and 10, the remaining 10 motifs are prevalent in most *ARF*s. *VaARF22*, *VaARF32*, *VaARF31-1*, and *VaARF31-2* contain only motif 1, and these four genes may be functionally redundant. *VaARF19-3* and *VaARF19-5* contain only motifs 8 and 10. Motif 10 is unique to *VaARF6*, *VaARF5-1*, *VaARF1-3*, *VaARF19-2*, *VaARF1-2*, *VaARF5-2*, *VaARF1-1*, *VaARF2*, *VaARF19-1*, *VaARF18-1*, and *VaARF18-2*. For the *IAA* gene family, all *VaIAA*s except *VaIAA27-1* contain motif 1, indicating that it plays an important and possibly structural role in *VaIAA*s.

### 2.5. Promoter and Synteny Analysis of VaARF-IAA Gene Families in Vaccinium bracteatum

Remarkably, every member of the *VaARF*-*IAA* gene families was found to possess light-responsive elements (Appendix A), highlighting the pivotal role of these genes in light-mediated physiological processes (Figure 6A,B). Similarly, hormone-responsive elements were ubiquitously present across all *VaARF* gene promoters, reflecting the gene family’s comprehensive involvement in hormonal signaling pathways. The majority of these promoters also featured MeJA-responsiveness, MYB binding sites indicative of drought inducibility, auxin responsiveness, and gibberellin-responsive elements, among others. Additionally, elements related to low temperature, defense (TC-rich repeats), and stress (CARE) were identified, signifying the gene family’s versatile role in plant growth, development, and stress response mechanisms.

Specifically, CARE elements, associated with stress-responsive elements, were uniquely identified in the promoters of *VaARF19-5* and *VaARF5-2*, while TC-rich repeats, signifying defense and stress responsiveness, were exclusively found in *VaIAA14-1* and *VaIAA27-3*. The diversity and specificity of cis-acting elements observed in the promoters of *VaARF*s and *VaIAA*s suggest a complex regulatory network influencing plant responses to environmental cues and hormonal signals.

### 2.6. Expression Analysis of VaARF-IAAs during Fruit Ripening

*VaARF2*, *VaARF6*, *VaARF4*, *VaARF5-1*, *VaARF1-2*, *VaARF19-5*, and *VaARF28* were highly expressed during the green fruit stage, suggesting a potential correlation with fruit enlargement (Figure 7A). Conversely, *VaARF5-2*, *VaARF18-2*, and *VaARF1-1* showed heightened expression in the blue fruit stage, which may be associated with the processes involved in fruit ripening.

Similarly, within the *IAA* gene family, 12 genes displayed significant differential expression across the fruit development stages. *VaIAA14-1*, *VaIAA27-4*, and *VaIAA27-3* were particularly active during the green fruit stage, further associating these genes with the process of fruit enlargement. On the other hand, *VaIAA1*, *VaIAA20*, and *VaIAA27-2* demonstrated peak expression levels in blue fruit, suggesting their roles in the maturation process.

### 2.7. Interaction Networks of VaARF-IAAs with Functional Genes

Auxins play an integral role in orchestrating plant growth and development. They regulate gene expression through specific transcription factors and proteins, which are fine-tuned to environmental responses within the signaling cascade [39]. Key among the auxin transcription factors are the auxin response factors (*ARF*s) and the *AUX/IAA* inhibitors. In pursuit of a deeper understanding of the biological functions and regulatory networks of *ARF*s and *AUX/IAA*s, we utilized a homology-based approach to predict protein–protein interactions (PPIs). Our study revealed the presence of several PPIs within the *ARF* and *AUX/IAA* gene families (Figure 8A), with *VaIAA1* and *VaIAA20* identified as the core components within these two gene families [40].

Furthermore, we analyzed the protein–protein interactions among *ARF*s, *AUX/IAA*s, and other functional genes (Figure 8B). As anticipated, the majority of the proteins interacting with *ARF* were validated as significant constituents of the auxin response pathway [41]. For instance, *ARF* interacts with PIN-FORMED (PIN) proteins, ATP-BINDING CASSETTE subfamily B (ABCB) proteins, MYB proteins, TIR proteins, and AFB proteins [42], all of which play vital roles in plant growth and development [43].

### 2.8. Protein 3D Structure Analysis

Three-dimensional models were generated using the primary sequences of VaARF and VaIAA proteins, which revealed distinct structural features that may correlate with functional specificity (Figure 9A,B). The structural variability observed among the VaARF proteins, such as differences in the orientation and configuration of secondary structural elements like α-helices and β-sheets, suggests specialized interactions and functions within the plant signaling pathways. The 3D structural models of key proteins such as *VaARF1-1*, *VaARF2*, *VaARF5-1*, and *VaARF19* series show notable differences in their domain architectures and surface properties. These variations may influence their binding affinities and interactions with other molecules, possibly affecting their role in auxin signaling and response mechanisms in *Vaccinium bracteatum*. Moreover, the analysis identified specific regions within these proteins that could potentially be involved in ligand binding or protein–protein interactions, critical for their function as transcriptional regulators. For example, the predicted models suggest that the *VaARF19-5* protein might have additional binding sites, which could be crucial for its interaction with other proteins or DNA elements within the plant cell.

## 3. Discussion

### 3.1. Interpretation of Phylogenetic Relationships and Gene Structure Diversity

In naming these identified genes, we adhered to a systematic approach by aligning them with their closest homologs in the NCBI database, thereby ensuring that the nomenclature reflects phylogenetic relationships and facilitates cross-species comparisons [44]. This methodology not only enhances the clarity of our findings but also aligns with standard practices in genomic studies, enabling researchers to easily correlate our identified genes with known auxin response mechanisms in other species [45].

The exhaustive identification and subsequent annotation of the *VaARF* and *VaIAA* gene families underscore the complexity of auxin signaling in *Vaccinium bracteatum.* These findings lay a crucial foundation for future functional studies aimed at unraveling the roles of these genes in growth, development, and stress responses in *Vaccinium bracteatum*, potentially opening new avenues for agricultural innovation and crop improvement.

*Vaccinium bracteatum*, a member of the Rhododendron family, is a diploid plant species characterized by 12 distinct chromosomes, as revealed in our previous research. The distribution of *ARF* and *IAA* genes across these chromosomes is notably uneven, highlighting the complex genetic architecture of auxin response mechanisms in this species. Chromosome 9 stands out with the highest gene distribution density for *VaARF* genes, whereas chromosome 5 leads in density for *VaIAA* genes. This uneven gene distribution and evidence of gene clustering and duplication provide valuable insights into the genomic organization and evolutionary dynamics of the *ARF* and *IAA* gene families in *Vaccinium bracteatum*. Such findings not only enrich our understanding of the genetic basis of auxin response in this species but also underscore the intricate relationship between chromosomal architecture and gene function diversification in plant genomes [46].

The arrangement of *VaIAA1*, *VaIAA27-1*, and *VaIAA26* in close locations, along with *VaIAA27-4* and *VaIAA18*, paired with the clustered positioning of *VaARF* genes including *VaARF18-2* and *VaARF4*, *VaARF31-1* and *VaARF31-2*, *VaARF19-2* and *VaARF19-3*, and *VaARF9* and *VaARF3*, strongly suggests the history of gene duplication events. These clusters may indicate regions of the genome that have undergone evolutionary expansions [47], potentially contributing to the diversification of the *ARF* and *IAA* gene functions in *Vaccinium bracteatum*.

The chromosomal collinearity analysis not only elucidated the structural organization and evolutionary dynamics of the *ARF* and *IAA* gene families in *Vaccinium bracteatum* but also provided insights into the mechanisms driving gene family expansion and functional conservation. This detailed mapping and comparison help pave the way for further functional genomic studies aimed at understanding the correlations between these gene families in plant development and stress responses [48].

### 3.2. Role of ARF Genes in the Fruit Ripening of Vaccinium bracteatum Based on Expression Profiles

The expression analysis of auxin response factor (*ARF*) genes during fruit ripening in *Vaccinium bracteatum* was meticulously carried out using transcriptome data from four developmental stages including leaves, green fruit, red fruit, and blue fruit, with green and red fruits representing unripe stages. This comprehensive dataset, derived from triplicate samples for each stage, provided a robust foundation for exploring the dynamic expression of *ARF* genes across different phases of fruit development [49].

Expression levels were quantified using FPKM (Fragments Per Kilobase of transcript per Million mapped reads) [50], enabling a precise measurement of gene activity. Our analysis revealed that 14 *ARF* genes exhibited significant differential expression throughout the fruit development process, suggesting a pivotal role for the majority of *ARF*-related genes in fruit maturation.

This detailed expression profiling of *ARF* and *IAA* genes underscores the complex regulatory network governing fruit development in *Vaccinium bracteatum* [51]. The distinct expression patterns observed not only highlight the specific contributions of individual *ARF* and *IAA* genes to fruit enlargement and ripening but also pave the way for future studies aimed at unraveling their functional roles in these critical developmental processes [52].

Further studies, including gene knockout experiments, are necessary to determine the definitive roles of these *ARF* genes in specific fruit developmental processes, such as cell division, growth, and maturation. Additionally, exploring the links between specific auxin sources, such as *YUCCA* gene expression, and the regulation of these *ARF* genes will provide deeper insights into the hormonal control mechanisms in fruit development.

### 3.3. Analysis of Interaction Networks

Auxin Polar Transport (PAT) is essential for auxin’s short-distance distribution, mediated by auxin carrier protein families within the cell membrane, including PIN-FORMED (PIN) proteins and ATP-BINDING CASSETTE subfamily B (ABCB) proteins [41]. The activity of these carrier proteins is influenced by the regulation of their expression and subcellular localization, which are modulated by *ARF* transcription factors in response to environmental signals.

In tomatoes, *SlPIN8* is specifically expressed in pollen, with its function possibly linked to pollen development [53] and auxin homeostasis [54]. TIR and AFB proteins are nuclear auxin-sensitive proteins containing nucleotide/leucine-rich repeats, allowing interaction with specific subsets of AUX/IAA proteins [54]. Research in *Arabidopsis* thaliana reveals that *TIR1/AFB* genes encode the primary auxin receptors that regulate plant growth, and *AFB3* mediates lateral root growth in response to nitrogen [55]. In soybeans, the *GmTIR1/AFB3* gene is a homolog of *TIR1* and *AFB3* found in *Arabidopsis thaliana*, serving to regulate soybean nodulation in response to rhizobia infection through the auxin signaling pathway [56]. In barley (*Hordeum vulgare*), the HvYUC protein is involved in mediating defense against salt stress, and a similar positive response to aluminum stress by *TAA1* through ethylene signaling has been observed [57]. The findings of this study significantly broaden our understanding of the regulatory networks of *ARF*s and *AUX/IAA*s, as well as their influence on plant growth and development [58].

### 3.4. Functional Implications of the Protein Structure Findings

The comprehensive phylogenetic analysis not only reveals the homologous relationships among *Vaccinium bracteatum ARF* genes, *IAA* genes, and those from other key plant species, but it also indicates a greater degree of evolutionary conservation within the *ARF* gene family compared with *IAA*s [59]. Such findings suggest that *ARF* genes may play a fundamental role in plant development and stress responses that have been preserved across divergent evolutionary paths [50]. The detailed subgrouping and cross-species comparisons provide valuable insights into the functional diversification and evolutionary history of the *ARF* and *IAA* gene families, offering a foundation for further functional characterization and exploration of their roles in plant biology [47].

The structural analysis of *VaARF* and *VaIAA* gene families in *Vaccinium bracteatum* reveals a remarkable diversity and complexity within their gene structures. A significant structural variance is observed between the *ARF* and *IAA* genes [60], where *ARF*s exhibit a higher exon/intron ratio compared with *IAA*s, suggesting a higher degree of complexity and diversity within the *ARF* family. This structural distinction could be indicative of the varied functional roles that these gene families play within plant developmental processes and stress responses. The elucidation of gene structures and domain compositions not only provides insights into the functional complexities of the *VaARF* and *VaIAA* gene families but also underscores the evolutionary strategies employed by *Vaccinium bracteatum* in modulating auxin signaling pathways. These findings pave the way for future functional characterization studies, aiming to unravel the specific roles of these gene families in plant growth, development, and adaptation to environmental cues.

In our comprehensive analysis of the promoter regions, spanning 2 kb upstream of the transcription start sites of each *VaARF* gene, we aimed to elucidate the potential regulatory mechanisms governing the *VaARF*-*IAA* gene family’s activity in *Vaccinium bracteatum*. This analysis uncovered a plethora of cis-acting elements predominantly associated with various plant hormone responses, underscoring the intricate relationship between the *VaARF*-*IAA* gene family and hormonal regulation [61].

This promoter analysis not only sheds light on the potential functional diversity within the *VaARF*-*IAA* gene family but also provides a foundation for future studies aimed at unraveling the molecular mechanisms by which these genes modulate plant growth and stress resilience in *Vaccinium bracteatum*. The identification of unique and common regulatory elements across these promoters underscores the multifaceted roles of *ARF* and *IAA* genes in orchestrating plant development and environmental adaptation [62].

To better understand the structural basis of ARF protein function in *Vaccinium bracteatum*, we utilized the Swiss Model web-based platform to predict and analyze the 3D structures of the ARF protein family. This analysis revealed substantial structural diversity among the ARF proteins [63], reflecting their varied roles in plant physiology and development. This comprehensive structural analysis not only enhances our understanding of the functional diversity of the *ARF* gene family in *Vaccinium bracteatum* but also provides a foundation for future experimental studies aimed at validating these models and elucidating the specific roles of these proteins in plant growth and development processes [64,65].

## 4. Materials and Methods

### 4.1. Plant Resources

Samples of *Vaccinium bracteatum* were collected from the hilly subtropical region in the southern part of Anhui Province (30°51′ N, 118°23′ E). This collection enabled us to obtain essential genomic data for the species. Complete genomic data and amino acid sequences of two gene families from *Arabidopsis thaliana*, *Glycine max*, *Brassica napus*, and *Gossypium hirsutum* L. were procured respectively from the TAIR website (https://www.arabidopsis.org/) (accessed on 25 September 2023), NCBI database (https://www.ncbi.nlm.nih.gov/) (accessed on 25 September 2023) and uniport database (https://www.uniprot.org/), (accessed on 25 September 2023). Leaves and fruits were collected from three separate developmental stages including the green fruit growth phase (Stage 1), pink fruit color accumulation phase (Stage 2), and mature blue fruit phase (Stage 3). Each developmental stage was sampled from three biological replicates. The samples were flash-frozen in liquid nitrogen followed by preservation at −80 °C until RNA extraction. The resultant RNA samples were thereafter utilized for gene prediction based on the transcriptome.

### 4.2. Genome-Wide Identification and Classification of ARF-Aux/IAA Gene Families

Information on *Arabidopsis thaliana ARF* and *AUX/IAA* gene families was downloaded from the TAIR database (https://www.arabidopsis.org/) (accessed on 25 September 2023). The JGI Plant Site (https://phytozome.jgi.doe.gov/pz/portal.html) (accessed on 27 September 2023) was then utilized to compare the homology of amino acids linked with *Arabidopsis thaliana* ARF and AUX/IAA proteins. Furthermore, the characteristics of *ARF* and *AUX/IAA* genes were identified through the NCBI database’s Blastp feature, followed by assigning specific names. The online ExPASy tool (https://www.expasy.org/) (accessed on 12 October 2023) was then utilized to obtain data on the biochemical qualities of each ARF and AUX/IAA protein, including parameters such as the number of amino acids, isoelectric point (pI), and molecular weight.

### 4.3. Visualization and Collinearity Analysis of Chromosomal Positioning

The chromosomal localization data for the verified *VaARF*-*IAA* gene family stems from the sequencing results obtained from *Vaccinium bracteatum*. Chromosome maps were illustrated using MG2C (http://mg2c.iask.in/mg2c_v2.1/) (accessed on 15 October 2023). To analyze and characterize the collinearity of the *ARF*-*AUX/IAA* gene families, TBtools (version 2.019) were utilized. These tools helped in identifying homologous and paralogous genes in the replication events of the *ARF* and *AUX/IAA* gene families. For the computation of the Ka/Ks ratio, which represents the rate of non-synonymous to synonymous substitutions among respective gene pairs, TBtools (version 2.019) was used. In addition, EMBL Pairwise Sequence Alignment (https://www.ebi.ac.uk/jdispatcher/psa/lalign) (accessed on 15 October 2023) was employed for carrying out a bidirectional sequence alignment, assisting in determining nucleotide level congruity within gene pairs.

### 4.4. The Establishment of the Phylogenetic Tree

By utilizing the tool known as “Find Best DNA/Protein Models (ML)” found in Molecular Evolutionary Genetics Analysis (version MEGA 11.0), optimal model tests were conducted on the amino acid sequences of ARF and AUX/IAA proteins. This ultimately facilitated the construction of a maximum likelihood model. Each tree node was computed utilizing 1000 bootstrap duplicates, with the remainder of the parameters set at their default values.

### 4.5. Analysis of Introns, Exons, and Regulatory Sequences

To illustrate the exon–intron structure diagram and the coding sequence (CDS) configuration of the *VaARF*-*IAA* genes, we leveraged the GENE Structure Display (GSDS 2.0) web tool (accessible at http://gsds.gao-lab.org/) (accessed on 9 January 2024). This was performed by uploading the genomic sequence and the coding sequence to create the gene structure. We analyzed the conservative base sequences of the ARF and AUX/IAA proteins in VaARF-IAAs using the online tool MEME Suite (available at https://meme-suite.org/meme/) (accessed on 20 January 2024). The base sequences found in the generated protein dataset were saved for future reference. For the visualization process, we employed TBtools software (version 2.019).

### 4.6. Analysis of Promoter Sequence

The promoter sequence analysis involved 2000 bp of the genome sequence upstream as the transcription initiation site. The software TBtools (version 2.019), was utilized to extract the 2000 bp upstream sequence of the *ARF* and *AUX/IAA* gene families, serving as the promoter sequence. Further analysis of the operative elements in the promoter region was conducted via the PlantCARE website (http://bioinformatics.psb.ugent.be/webtools/plantcare/html/) (accessed on 5 March 2024). Upon generating the data, it was prudently organized and simplified. The visualization process was facilitated using TBtools (version 2.019) software.

### 4.7. Expression Patterns within Different Fruit Stages 

Total RNA was isolated from 12 samples, precisely from tissues of leaves and green, pink, and blue fruits, with each tissue represented by triplicates. The data depicted in the figure are the products of a transcriptome analysis used to examine tissue specificity and the differential expression of *ARF* and *AUX/IAA* gene families across varying regions within the plant.

The mapping of these RNA-seq reads was completed using TBtools (version 2.019), employing its default settings. Quantitative gene expression analysis was conducted using the TPM (Transcripts Per Million) algorithm. Additionally, the heatmap of the resultant TPM values for *ARF* and *AUX/IAA* from each tissue was generated using the heatmap function of TBtools (version 2.019).

### 4.8. Analysis of Protein–Protein Interaction Expression

Utilizing the AraNetV2 (http://www.inetbio.org/aranet/) (accessed on 12 March 2024) and STRING (http://string-db.org/) (accessed on 15 March 2024) databases, we established the interaction networks between *VaARF* and *VaIAA*s. To visualize these predicted interaction networks, we employed Cytoscape (version 3.7.2). In addition, the three-dimensional structural homology of the *VaARF*-*IAA* gene family’s spatial protein models in Vitex agnus-castus (Chaste tree) was modeled using 3D structure models provided by ExPaSy (https://swissmodel.expasy.org/interactive) (accessed on 26 March 2024).

## 5. Conclusions

In our research, we identified 26 *ARF* genes and 17 *AUX-IAA* genes within the genome of *Vaccinium bracteatum*. We conducted an in-depth analysis of their phylogeny, gene structure, conserved domains, and motifs to ascertain their evolutionary affiliations with other plant species. The gene expression profiles during different stages of *Vaccinium bracteatum* fruit development revealed a spectrum of expression diversity in the *ARF* and *IAA* gene families. Furthermore, interaction analyses of *VaARF* and VaAUX-IAA with various functional genes unveiled their critical role in plant growth and development. These findings lay a solid foundation for future research into the operational characterization of *ARF* genes, *AUX-IAA* genes, and *ARF*-mediated signal transduction pathways.

## Figures and Tables

**Figure 1 ijms-25-08385-f001:**
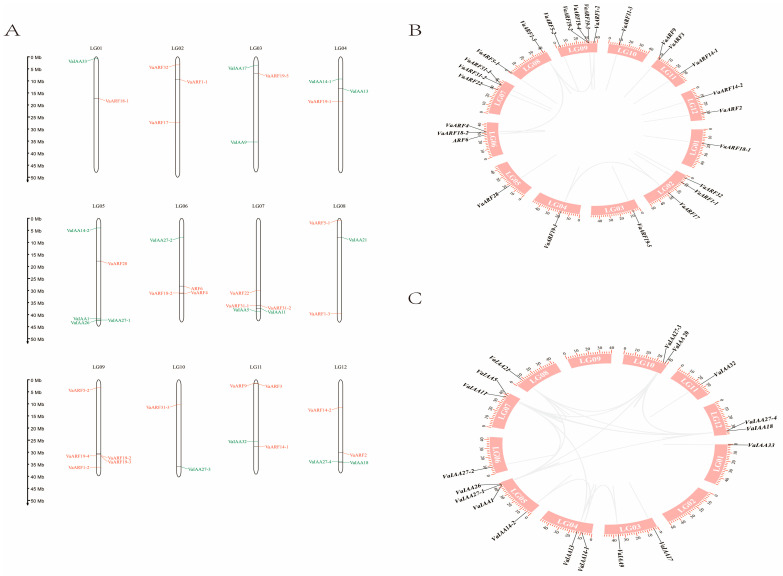
The distribution of *VaARF* and *VaIAA* family members on the chromosomes of *Vaccinium bracteatum*. (**A**) The chromosome mapping of *VaARF*s-IAAs and (**B**) collinearity among *VaARF* gene families within the chromosome. (**C**) collinearity among *VaIAA* gene families within the chromosome.The chromosome number is listed at the top of each vertical bar. The ruler depicts the length of the chromosome/Mb. The red color on the chromosome is the *ARF* gene, and the green color is the *IAA* gene. The gray line indicates that most of the *ARF* and *IAA* genes have a collinear relationship among species. Chromosomes 1–12 are shown in pink, with *ARF* and *IAA* gene names on the periphery.

**Figure 2 ijms-25-08385-f002:**
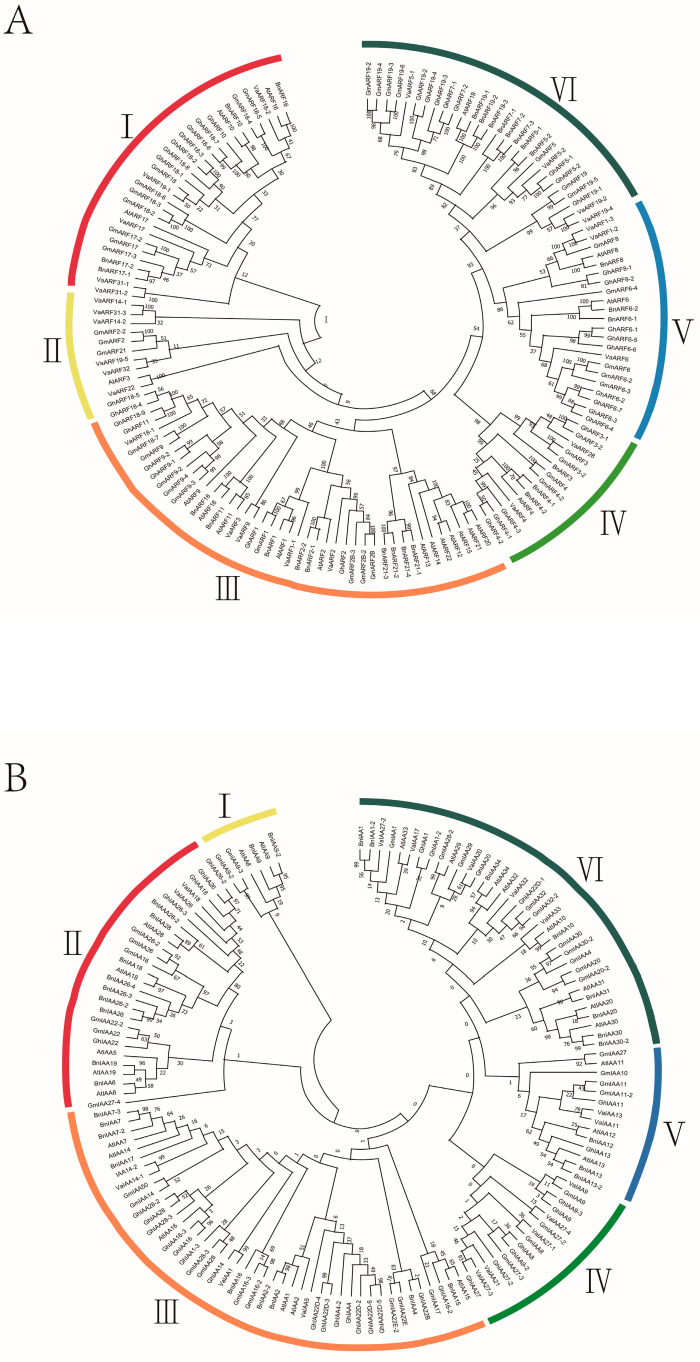
Phylogenetic relationships among *VaARF*, *VaIAA*, and five other families of species. (**A**) Phylogenetic analysis of *VaARF* (**B**) Phylogenetic analysis of *VaIAA*. The phylogenetic tree was established utilizing the full-length gene sequence of *VaARF*-*IAA*s from *Vaccinium bracteatum* and *Arabidopsis thaliana*, *Glycine max*, *Brassica napus*, and *Gossypium hirsutum* L.

**Figure 3 ijms-25-08385-f003:**
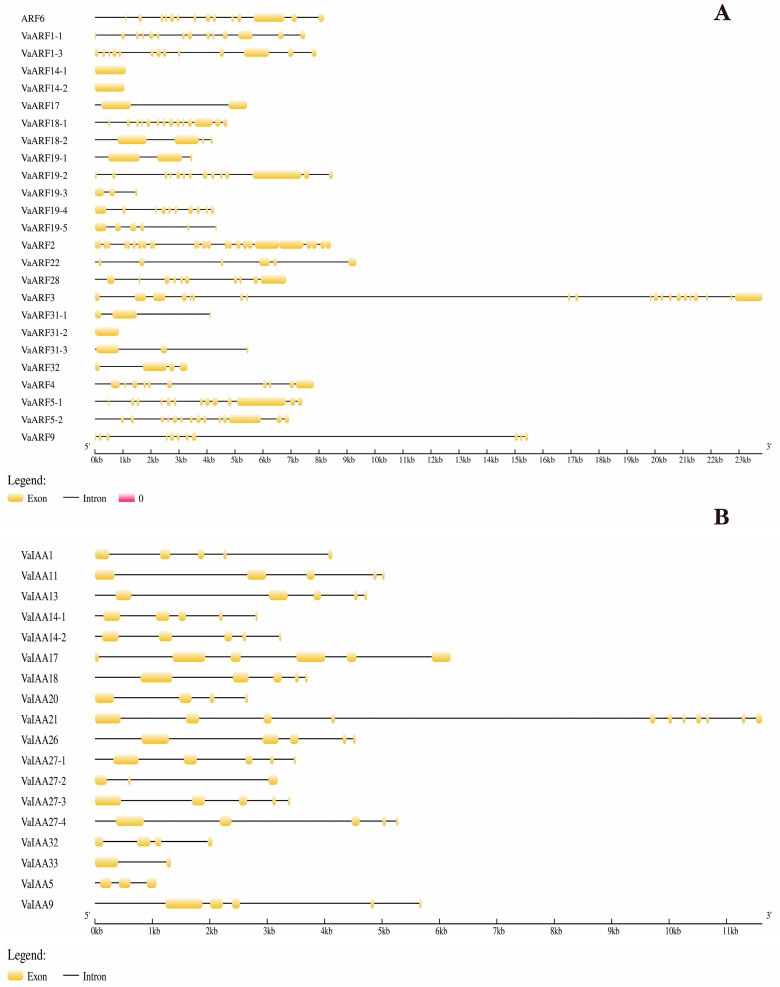
Genetic structure analysis of *VaARF* and *VaIAA* gene families. (**A**) Genetic structure analysis of *VaARF* (**B**) Genetic structure analysis of *VaIAA*. Exons and introns are represented by yellow boxes and black lines, respectively.

**Figure 4 ijms-25-08385-f004:**
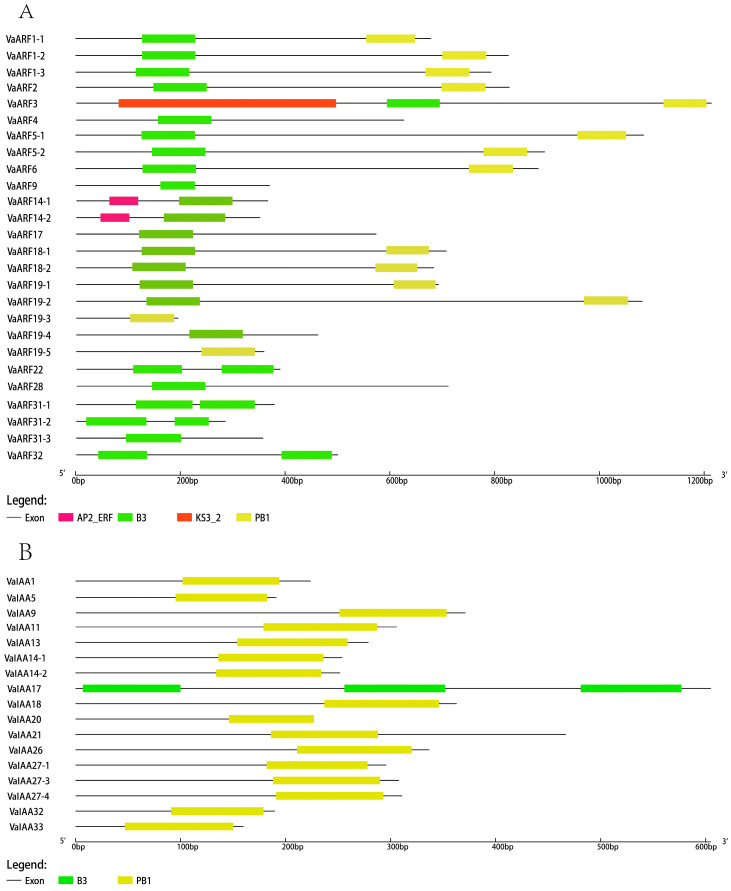
Domain analysis of *VaARF* and *VaIAA* gene families. (**A**) Domain analysis of *VaARF*s (**B**) Domain analysis of *VaIAA*s. Different colors represent different domains.

**Figure 5 ijms-25-08385-f005:**
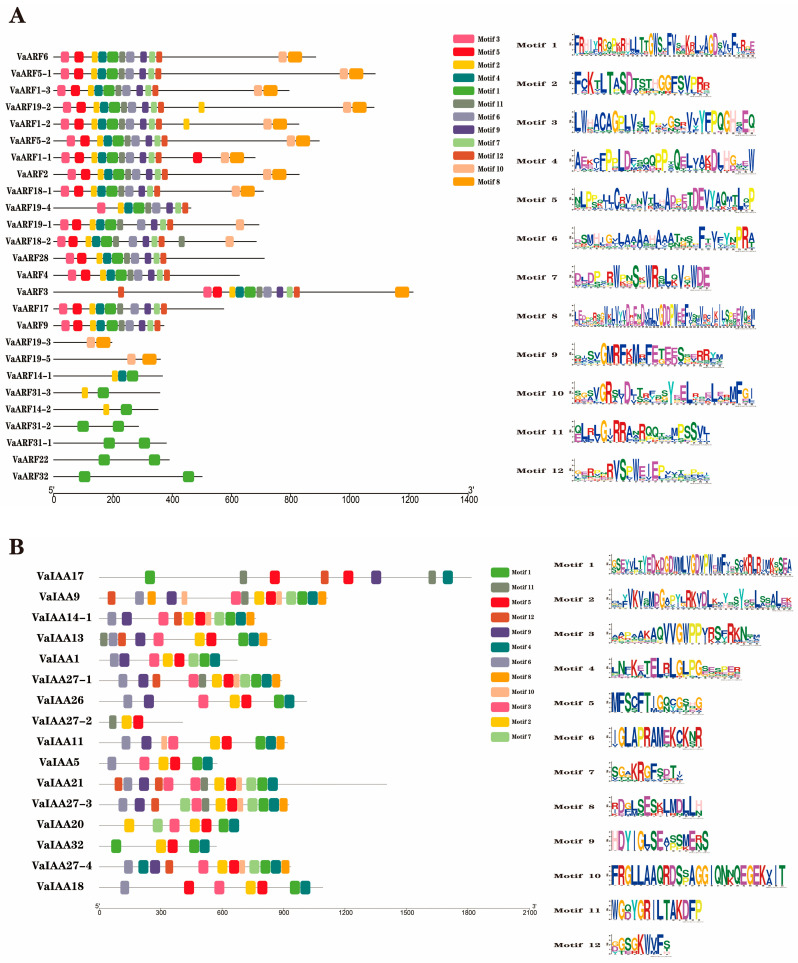
Conserved motifs of the *VaARF* and *VaIAA* gene families. (**A**) Conserved motifs of the *VaARF*s (**B**) Conserved motifs of the *VaIAA*s. The motif length can be calculated using the ruler at the bottom. The ten expected motifs are represented by colored boxes on the left, with the Motif LOGO on the right.

**Figure 6 ijms-25-08385-f006:**
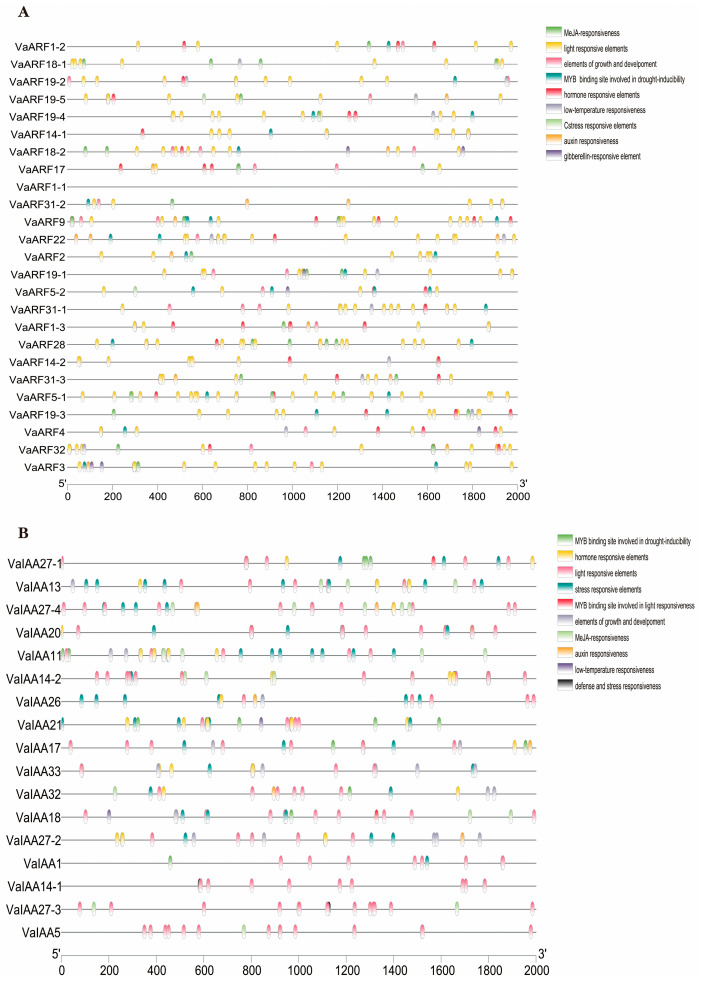
Analysis of cis-regulatory elements in the *VaARF* and *VaIAA* families. (**A**) Analysis of cis-regulatory elements in the *VaARF*s (**B**) Analysis of cis-regulatory elements in the *VaIAA*s. Trans-acting regulatory elements discovered in the promoter regions are represented in different blocks colored according to the type of element.

**Figure 7 ijms-25-08385-f007:**
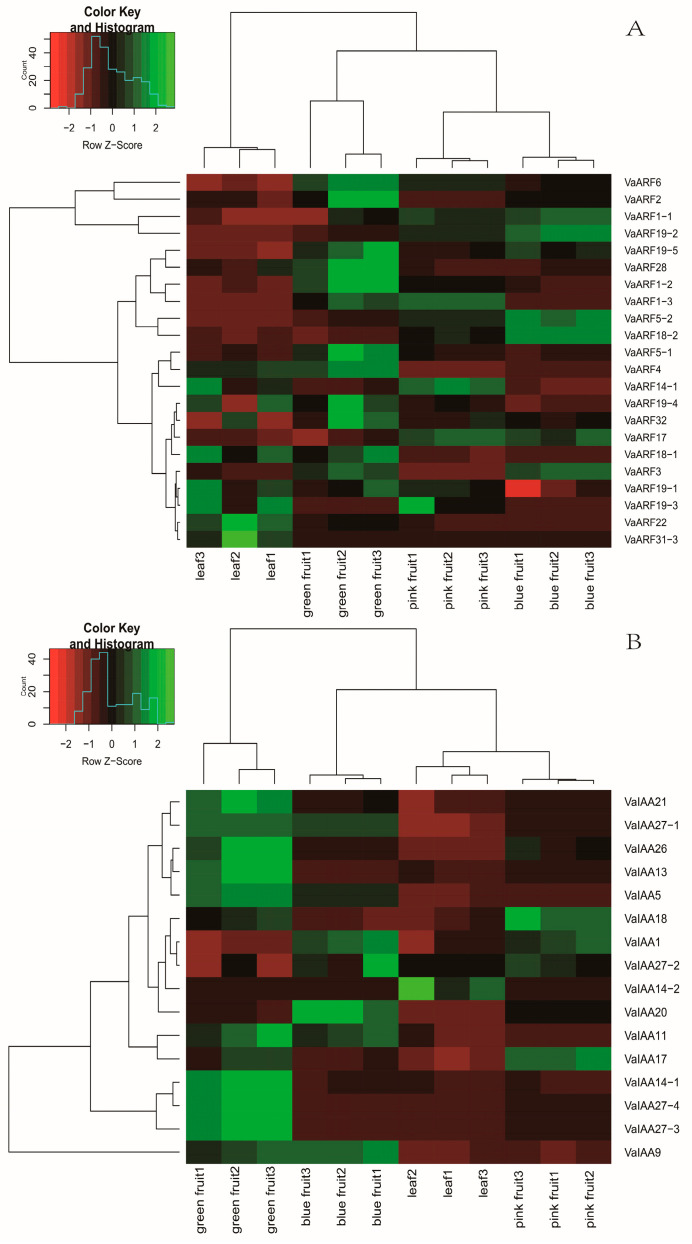
Expression pattern in the *VaARF* and *VaIAA* gene families during different stages of fruiting. (**A**) Expression pattern in the *VaARF*s (**B**) Expression pattern in the *VaIAA*s.

**Figure 8 ijms-25-08385-f008:**
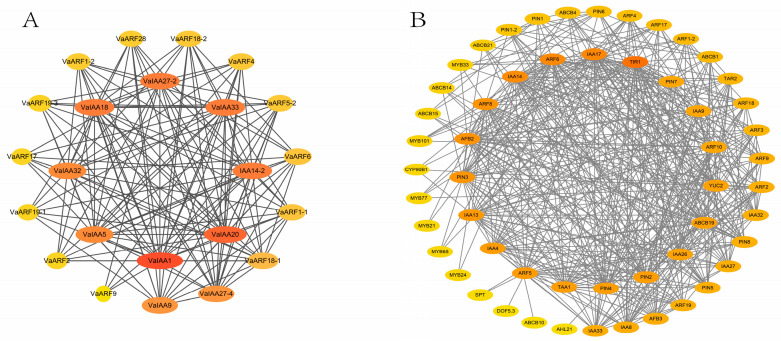
Interaction protein analysis of the *VaARF* and *VaIAA* gene families. (**A**) The interaction between the two gene families. (**B**) Prediction of protein–protein interactions between *ARF* protein and other functional proteins. The dark color of the circles and lines represents a high degree of interaction among proteins. The size and dark color of the circles of each protein show high interactions with other proteins.

**Figure 9 ijms-25-08385-f009:**
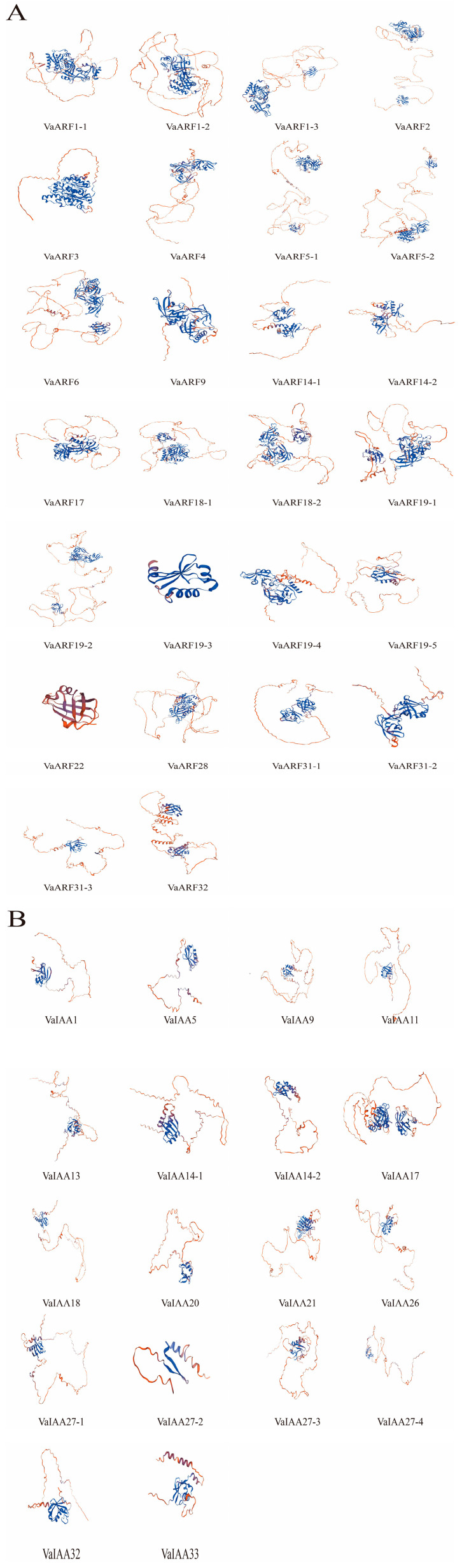
Three-dimensional structure model of VaARF/VaIAA proteins. (**A**) Three-dimensional structure model of VaARF proteins (**B**) Three-dimensional structure model of VaIAA proteins. Ramachandran plots were also used to validate the models of the various proteins that were obtained.

## Data Availability

The original contributions presented in the study are included in the article/Appendix A, further inquiries can be directed to the corresponding authors.

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
