# Peer review of "Genome-Wide Identification, Expression, and Interaction Analysis of the Auxin Response Factor and AUX/IAA Gene Families in Vaccinium bracteatum"

_ijms, 2024, doi:10.3390/ijms25158385_

Round 1

Reviewer 1 Report

Comments and Suggestions for Authors

The paper described análisis of AUX/IAA and ARF gene families in Vaccinium bracterium.

The authors performed dettailed análisis of genoma, indentified ARF and Aux/IAA gene members and classified it for new species.

The data are interesting and important and allow us to better understand mechanism of growth regulation in this plants.

However, the presentation need to be definitely imporoved.

Line 13: vital – redundant.

Lines 26 – 27: You mentioned only one gene family here!

Line 37: it will be nice to mention that each auxin-reguklated process were linked with cell-type specific auxin source and mention briefly such sources like 11 YUCCAs in Arabidopsis, 12 YUCCA genes in tomato etc.

Lines 65- 66: it will be nice to mention cell-type specifci of ARF-Aux/IAA expression and link it with different local auxin sources. At least mention this point, indeed!

Line 91: “provide high antocyan content” ??? May be have?

Line 107: “resulting in essential genomic data for the species” ¿??

Line 119: harvested = download.

Figures: please, try to insert in pdf figures with at least 300 dpi, better 600 dpi. Now it is look like low resolution. And improve layoout. On Fig 2 B III cross line, etc.

Line 196: “Chromosome 9 boasts the highest concentration” ¿ Concentration¿??

Line 321: I would suggest to tell about correlation of gene expression level and fruit growth. For the involvemnet more details study with gene knockout expression were required.

Moreover, it will be nice in future clarify specific processes regulated by these genes (cell division, growth, cell type etc) and linked with specifci auxin source (YUCCAs expression etc).

Line 352: “These genes are regulated by gene expression” ¿???

Please clarify this part and explain your thoughts more clear.

Fig 9: too smal and not clear image. See above about resolution.

Line 418: not role, but correlation so far.

Comments on the Quality of English Language

moderate polishing are required

Author Response

Comments 1: Line 13: vital – redundant.

Response 1:Thank you for your suggestion.We have removed the word vital from the article.

Comments 2: Lines 26 – 27: You mentioned only one gene family here!

Response 2: Thank you for your attention to detail. In my manuscript, when I refer to VaIAA1 and VaIAA20 as core members, I intend to emphasize that these two proteins are pivotal in the interactions within and possibly between these families, given their extensive interactions observed in our analysis. I apologize for any confusion caused by the phrasing and will clarify this in the revised manuscript by stating: "Based on protein interaction predictions, VaIAA1 and VaIAA20 were designated as core members of their respective gene families, ARF and IAA.”

Comments 3: Line 37: it will be nice to mention that each auxin-reguklated process were linked with cell-type specific auxin source and mention briefly such sources like 11 YUCCAs in Arabidopsis, 12 YUCCA genes in tomato etc.

Response 3: Thank you for your valuable suggestion to elaborate on cell-type specific auxin sources and mention specific examples such as the YUCCA genes in different species. I have added relevant information to the manuscript to highlight the diversity and role of YUCCA genes in auxin biosynthesis across different plant species.

We have revised the article as follows: “It plays a central role in numerous biological processes including the development of various organs, regulation of shoot apical dominance, and differentiation of conducting tissues like xylem and phloem(Ali, Wang et al. 2022). In Arabidopsis, YUCCA2 (YUC2) regulates chloroplast RNA editing by modulating auxin levels, affecting the expression of editing factors through ARF1-dependent pathways (Li, Li et al. 2023). Genome-wide analysis in Mikania micrantha identified 18 YUCCA genes with diverse expression patterns and significant roles in plant growth and auxin biosynthesis (Luo, Xiao et al. 2022).”

Comments 4: Lines 65- 66: it will be nice to mention cell-type specifci of ARF-Aux/IAA expression and link it with different local auxin sources. At least mention this point, indeed!

Response 4: Thank you for your insightful suggestion to detail the cell-type specificity of ARF-Aux/IAA expression and its connection with local auxin sources. I agree that highlighting these aspects will significantly enrich the discussion of auxin signaling dynamics within the manuscript. In response to your comment, I have added some references in the manuscript as follow:

“Previous studies have highlighted the complexity of auxin signaling, noting that the specificity of ARF and Aux/IAA interactions can lead to diverse developmental and physiological outcomes. The cell-type specificity of ARF-Aux/IAA expression in relation to different local auxin sources is a critical area of study in plant development. Research highlights how distinct cell types interpret auxin signals to mediate various developmental processes. Rademacher(Rademacher, Möller et al. 2011) provided a cellular expression map of ARF genes, showing complex overlapping patterns during embryogenesis and primary root meristem development. This study underscores the developmental specificity generated by transcriptional regulation of ARF genes. Bargmann and Estelle (Bargmann and Estelle 2014) speculated on the tissue specificity of auxin responses through the SCF(TIR1)-Aux/IAA-ARF pathway, addressing how different cell types respond to the same auxin gradient. Ding (Ding, Zhang et al. 2021) mapped the tissue-specific contributions of auxin signals in root growth, demonstrating significant roles of specific cell types like the endodermis.”

Comments 5: Line 91: “provide high antocyan content” ??? May be have?

Response 5: Thank you for pointing out the ambiguous phrasing regarding the anthocyanin content in Vaccinium bracteatum Thunb. You are correct that the expression "provide high anthocyanin content" could be misinterpreted. My intention was to convey that the fruits of Vaccinium bracteatum are naturally rich in anthocyanins. To clarify this, I propose revising the sentence to:

"Importantly, Vaccinium bracteatum Thunb., an Ericaceae plant, is known for its berries that turn purple-black when ripe, bearing flowers in June-July and fruits in August-October. These small, deep blue fruits are naturally high in anthocyanin content, making this species a valuable ornamental fruit-greening plant."

Comments 6: Line 107: “resulting in essential genomic data for the species” ¿??

Response 6: Thank you for your comment concerning the phrase "resulting in essential genomic data for the species". I realize that the causal relationship implied by "resulting" may not be clearly articulated in the sentence. To clarify, my intention was to express that the collection of samples from the hilly subtropical region in the southern part of Anhui Province provided an opportunity to gather crucial genomic data for Vaccinium bracteatum.

To improve clarity and accuracy, I propose revising the sentence to:

"The samples of Vaccinium bracteatum were collected from the hilly subtropical region in the southern part of Anhui Province (30°51'N, 118°23'E). This collection has enabled us to obtain essential genomic data for the species."

Comments 7: Line 119: harvested = download.

Response 7: Thank you for your suggestion. To be more precise, we will replace the term "download" with "harvested " in the article.

Comments 8: Figures: please, try to insert in pdf figures with at least 300 dpi, better 600 dpi. Now it is look like low resolution. And improve layout. On Fig 2 B III cross line, etc.

Response 8: Thank you for your valuable advice. In order to improve the viewing of the pictures, we have improved the clarity of the pictures to ensure that the clarity of each picture is above 300 dpi. The picture layout has been modified in Figure 2 to ensure that there is no crossover between the picture content and the picture identity.

Comments 9: Line 196: “Chromosome 9 boasts the highest concentration” ¿ Concentration¿??

Response 9: Thank you for your query regarding the use of the term "concentration" in describing the distribution of VaARF genes on Chromosome 9. I appreciate your attention to detail and agree that the term could be misinterpreted. To better convey the intended meaning, I propose revising the sentence to:

"Chromosome 9 has the highest density of VaARF genes, with a total of five identified, whereas chromosomes 1, 3, 4, 5, and 10 each harbor a single VaARF gene."

Comments 10: Line 321: I would suggest to tell about correlation of gene expression level and fruit growth. For the involvemnet more details study with gene knockout expression were required.

Moreover, it will be nice in future clarify specific processes regulated by these genes (cell division, growth, cell type etc) and linked with specifci auxin source (YUCCAs expression etc).

Response 10: Thank you for your insightful comments regarding the correlation between gene expression levels and fruit growth stages in our study. I appreciate your suggestions for providing a more precise description of the relationships and for considering future studies involving gene knockout experiments to establish causality.

In response to your feedback, I propose to modify the text to better reflect the correlational nature of our findings, rather than implying direct causality. The revised sentence will read:

"VaARF2, VaARF6, VaARF4, VaARF5-1, VaARF1-2, VaARF19-5, and VaARF28 were highly expressed during the green fruit stage, suggesting a potential correlation with fruit enlargement. Conversely, VaARF5-2, VaARF18-2, and VaARF1-1 showed heightened expression in the blue fruit stage, which may be associated with the processes involved in fruit ripening."

Additionally, I have added a statement to discussion section 4.2 as you suggested:

"Further studies, including gene knockout experiments, are necessary to definitively determine the roles of these ARF genes in specific fruit developmental processes, such as cell division, growth, and maturation. Additionally, exploring the links between specific auxin sources, such as YUCCA gene expression, and the regulation of these ARF genes will provide deeper insights into the hormonal control mechanisms in fruit development."

Comments 11: Line 352: “These genes are regulated by gene expression” ¿??? Please clarify this part and explain your thoughts more clear.

Response 11: Thank you for your comment regarding the phrase “These genes are regulated by gene expression” in line 352. I realize that this statement was unclear and did not accurately convey the regulatory mechanisms I intended to describe.

To clarify, my intention was to emphasize that the activity of auxin carrier proteins such as PIN and ABCB is influenced by the regulation of their gene expression and subcellular localization, which are modulated by ARF transcription factors in response to environmental signals. I propose the following revision for better clarity:

"Auxin Polar Transport (PAT) is essential for auxin's short-distance distribution, mediated by auxin carrier protein families within the cell membrane, including PIN-FORMED (PIN) proteins and ATP-BINDING CASSETTE subfamily B (ABCB) proteins. The activity of these carrier proteins is influenced by the regulation of their expression and subcellular localization, which are modulated by ARF transcription factors in response to environmental signals."

Comments 12: Fig 9: too small and not clear image. See above about resolution.

Response 12: Thank you for your advice. For ease of reading, we used higher definition pictures as illustrations.

Comments 13: Line 418: not role, but correlation so far.

Response 13: Thank you for your comments on the role of these gene families in line 418. Our intention is to understand the relevance of these gene families in plant development and stress response. So we've changed it to the correlation of these gene families.

Reviewer 2 Report

Comments and Suggestions for Authors

The manuscript “Genome-Wide Identification, Expression and Interaction Analysis of ARF and AUX/IAA Gene Family in Vaccinium bracteatum”comprehensively studies the auxin gene family including ARF and Aux/IAA in plant fruit, Vaccinium bracteatum. Gene structure, phylogenetics relationship and expression profiling during fruit ripening were revealed in this study. Further, protein interaction networks of ARF and Aux/IAA and 3D protein structure were displayed in this work. 

Minor errors

Page 1, Line 19, 32, 130, Vaccinium bracteatum, please italicize the scientific word, please be consistent throughout the manuscript

Page 5, Figure 1 B and C showing the chromosomal collinearity is distorted, please adjust accordingly

Page 12, Figure 7, Different stages of fruit ripening should be labeled in English (eg. qingguo, hongguo, and languo) 

Page 13, Line 368, reference, please clarify

Page 15, Line 445, Arabidopsis thaliana, please italicize the scientific word

Page 14, Figure 9 A and B are in low resolution, please adjust it so that three dimensional structure can be seen by reader

Page 17, Line 509, author contribution, repeated word, please drop one of them

Page 17-20, Reference number from 1 to 61 were repeated twice in the manuscript

Other concerns

  1. Through the 3-D structure of the protein, did the author validate or verify the interaction network of VaARF-IAA with any of the transcription factors?

  2. Did the authors check the subcellular localization of the protein ARF and Aux/IAA?

Author Response

Comments 1: Page 1, Line 19, 32, 130, Vaccinium bracteatum, please italicize the scientific word, please be consistent throughout the manuscript.

Response 1: Thank you for reading this article carefully, we have put lines 19, 32, 130 Vaccinium brteatum, species names in italics, and checked the full article to make sure there are no other species name formatting issues.

Comments 2: Figure 1 B and C showing the chromosomal collinearity is distorted, please adjust accordingly.

Response 2: Thank you for your suggestion and we agree with that. Therefore, I reworked Figure 1 to show the collinearity between genes in a more obvious color. In order to prevent overlap between wires, we also adjusted the size ratio of the line segments.

Comments 3: Figure 7, Different stages of fruit ripening should be labeled in English (eg. qingguo, hongguo, and languo) .

Response 3: Thanks for your suggestion, we have revised the expression of the name of the four fruit development stages of Vaccinium bracteatum in Figure 7 to a more appropriate expression. The four stages are described as: leaf; green fruit ; pink fruit; blue fruit.

Comments 4: Line 368, reference, please clarify.

Response 4: Thank you for bringing to our attention the issue with the references in the manuscript. We have identified a problem of duplicate serial numbers in the reference section and have taken steps to rectify this issue.

Comments 5: Line 445, Arabidopsis thaliana, please italicize the scientific word.

Response 5: Thank you for reading this article carefully, we have put Arabidopsis thaliana, species names in italics, and checked the full article to make sure there are no other species name formatting issues.

Comments 6: Figure 9 A and B are in low resolution, please adjust it so that three dimensional structure can be seen by reader.

Response 6: Thank you for your advice. We used higher definition pictures as illustrations.

Comments 7: Line 509, author contribution, repeated word, please drop one of them

Response 7: Thanks for your suggestion, we have removed the duplicate content in the author contribution section of 509 lines.

Comments 8: Reference number from 1 to 61 were repeated twice in the manuscript

Response 8: Thank you for your reminding. We apologize for the repeated serial numbers in the reference section. We have modified and re-labeled them.

Comments 9: Through the 3-D structure of the protein, did the author validate or verify the interaction network of VaARF-IAA with any of the transcription factors?

Response 9: Thank you for your inquiry regarding the validation of the interaction network of VaARF-IAA with transcription factors using protein 3-D structure analysis. At present, our study did not include validation through protein structure analysis. However, this insightful suggestion opens up an exciting avenue for future research in our investigation.

Validating the interaction network of VaARF-IAA with transcription factors through protein structure analysis is a promising area for further exploration. This future direction aligns with the importance of understanding the molecular mechanisms underlying the regulatory interactions between these key components in auxin signaling pathways.

We appreciate your suggestion, and we plan to consider this aspect in our future research endeavors to deepen our understanding of the intricate network of regulatory interactions involving VaARF-IAA and transcription factors.

Thank you for highlighting this potential area for enhancement, which will undoubtedly strengthen the comprehensiveness of our study.

Comments 10: Did the authors check the subcellular localization of the protein ARF and Aux/IAA?

Response 10: Thank you for your question, we did not investigate the subcellular localization of the ARF and Aux/IAA proteins.

Round 2

Reviewer 1 Report

Comments and Suggestions for Authors

Thank you!

Everything is fine now, but except comment 2!

VaIAA1 and VaIAA20 both seems to be IAA family, not ARF and IAA.

Maybe you mean VaARF1? I means in this poinjt only correct name.

My best regards!

Author Response

Comments 1: VaIAA1 and VaIAA20 both seems to be IAA family, not ARF and IAA. Maybe you mean VaARF1? I means in this poinjt only correct name.

Response 1: Thank you for your suggestion. We confirm that the gene name is correct. According to the analysis results of fig8, among the two gene families, VaIAA1 and VaIAA20 interact most closely with other genes. Therefore, we speculated that these two genes are the core members of the IAA gene family and play an important role in the functioning of the two gene families. To avoid ambiguity, we adjust to your comments: “Based on protein interaction predictions, VaIAA1 and VaIAA20 were designated as core members of  VaIAA gene families.”
